# Military Object Detection Using a Fine-Tuned Florence-2 Vision Model

## Abstract

Accurately detecting military vehicles and equipment in real-world environments is a challenging but vital task in modern defense Liu et al. (2022); Jacob et al. (2023). In this work, we fine-tuned Microsoft's Florence-2 Large model Xiao et al. (2023) to recognize a wide range of military assets, including tanks, helicopters, artillery, and ground troops under realistic battlefield conditions. Instead of training on clean or staged images, we built a dataset of over 7,000 images collected from military exercises and surveillance footage. These images included difficult cases like camouflage, partial visibility, and low lighting. The objects were annotated using tools such as CVAT GitHub and Roboflow Roboflow, allowing us to maintain consistency across all categories. To improve performance on underrepresented classes, we used data augmentation Shorten & Khoshgoftaar (2019); Perez & Wang (2017) and class-aware sampling Buda et al. (2018). Our model achieved strong results, with a precision of 98.95%, recall of 99.33%, and an average IoU of 85.29%. These outcomes show that the model performs well even in messy, real-world conditions. This work highlights the potential of Florence-2 Large for practical defense applications like drone surveillance, battlefield monitoring, and automated threat detection.

## 1 Introduction

In today's fast-changing battlefield environment, the ability to detect and respond to threats quickly can make all the difference TTMS (2024). With this in mind, our work presents the first fine-tuned version of Microsoft's Florence-2 Large model Xiao et al. (2023), trained specifically to identify tanks, artillery, and fighter aircraft in real-world military settings. The goal is simple yet critical to help our forces on the ground spot potential enemy threats in time, especially those camouflaged or hidden in the heat of the battlefield Fan et al. (2020).

Unlike general-purpose models that are trained on clean, studio-like images, this project focuses on a much harder task: recognizing military equipment as it appears in actual combat zones, dusty terrains, dense forests, damaged structures, and sometimes even under poor lighting. Prior works such as Liu et al. (2022); Bahi et al. (2024) have highlighted the difficulty of object detection under such adverse conditions. Most datasets available online show tanks and jets in display conditions, clean, centered, and fully visible. But real war doesn't happen in exhibition grounds.

One of our biggest challenges was to find a model architecture capable of understanding such complex, real-world imagery. After comparing several leading models, we selected Microsoft's Florence-2 for its powerful combination of image understanding and language processing Xiao et al. (2023); Dosovitskiy et al. (2020). But even the best model needs the right data and that's where the real work began.

We built a custom dataset from scratch, collecting over 7,000 images of tanks, armored vehicles, and fighter planes, many from surveillance footage, battlefield simulations, and military drills Mundhenk et al. (2016). These images were then manually annotated, every tank, gun, and aircraft was marked by hand, to train the model on what to look for. Special care was taken to ensure that even camouflaged or partially hidden objects were included, reflecting the kind of real-time challenges faced by soldiers in the field Jacob et al. (2023); Li et al. (2017).

Training such a model was no easy task. Camouflage, shadows, damaged vehicles, and background clutter made it tough, but that was the point. We wanted the model to be tough too, able to perform not just in theory but on the ground Cai et al. (2024). With this work, we take a significant step toward building AI tools that are not just smart, but battle-smart. Tools that can one day support our armed forces in surveillance, reconnaissance, and automated threat detection, whether on drones, defense vehicles, or command centers Mittal et al. (2020); Fink et al. (2020).

## 2 DATASET

The dataset used in this project was fully self-curated, keeping in mind the need for real-world relevance. Instead of relying on standard datasets that often feature tanks and artillery in clean, staged environments, we went the extra mile to gather visuals that depict these military assets in actual battlefield conditions, that is dusty terrains, damaged structures, and fast-moving situations Mittal et al. (2020); TTMS (2024).

To do this, we carried out an extensive online search, sourcing images from a wide range of articles, archived footage, and frames extracted from videos that showcased real-time utilization of army utilities Shen et al. (2025). Our focus was on capturing tanks, artillery guns, and fighter aircraft during drills, field operations, and actual combat scenarios, rather than in exhibitions, showrooms, or parade grounds.

This approach helped us build a dataset that truly reflects the population data of how military vehicles appear when in actual battlefield, partially visible, camouflaged, or even damaged. It wasn't just about creating a model that works in ideal conditions, but one that understands how things look on the ground, in the middle of action Dosovitskiy et al. (2020).

Of course, with this kind of raw data, refinement was key. Several images turned out to be too blurry or unclear, in fact, some were hard to recognize even for the human eye. Acknowledging these limitations and to stay true to quality standards, we applied a human baseline filter and truncated the dataset accordingly. What remained was a carefully cleaned and well-annotated collection of 7000 images ready for fine-tuning the Florence-2 model Shorten & Khoshgoftaar (2019).

The dataset consisted of the following classes of military equipment: Tank, Airplane, Artillery, Soldiers, Infantry Fighting Vehicle (IFV), Armoured Personnel Carrier (APC), Engineering Vehicle, Helicopter.

Each class represents a distinct category of army utility, and all instances were manually labeled to ensure clarity and consistency during model training.

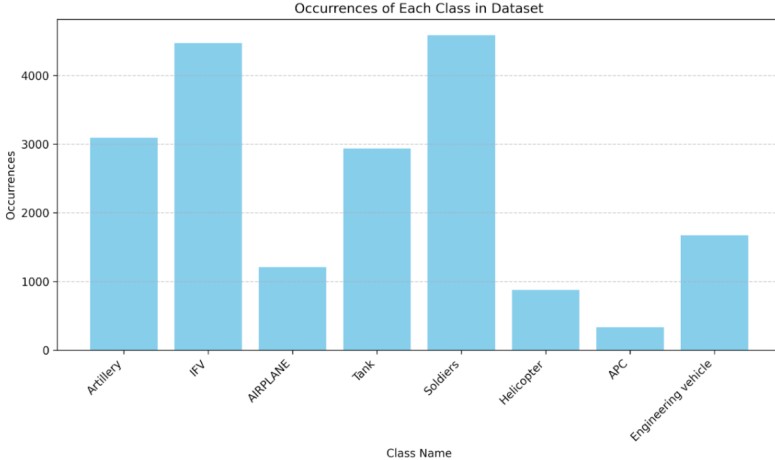

Figure 1: Class distribution of the dataset across different categories of military equipment (Tank, Airplane, Artillery, Soldiers, IFV, APC, Engineering Vehicle, Helicopter).

# 3 METHODOLOGY

Our main goal was always to build a system that works firmly in real battlefield conditions, not just in a room setup. Each step, from selecting the core model to identifying the end results, was designed for this practical purpose in mind.

## 3.1 DATA PREPARATION

The uncooked pictures we accrued varied plenty in great size, so we needed to lead them to steady.

**Cleaning & Resizing:** We made all pictures the equal size of 224x224, so our model was given a uniform centre Roboflow.

**Quality Check:** We tossed out any photographs that had been remarkably blurry or distorted – the type even someone would wear to make sense of.

**Normalization:** We adjusted the pixel values to reduce variations in lighting fixtures and assessment. This helped the version awareness of the real gadgets, no longer just how brilliant or darkish something was Buda et al. (2018).

## 3.2 ANNOTATION PROCESS

Since we constructed this dataset ourselves, each individual item in every image required careful annotation. We used tools like CVAT and Roboflow to draw bounding boxes around eight distinct categories: Tank, Airplane, Artillery, Soldiers, Infantry Fighting Vehicle (IFV), Armoured Personnel Carrier (APC), Engineering Vehicle, and Helicopter GitHub; Roboflow. To ensure accuracy, every label was always double-checked by another annotator. The final dataset was stored in the .jsonl (JSON Lines) format. A snippet of the dataset structure is shown below:

```
{"prefix": "<PREFIX>", "suffix": "<ENTITY><LOC_#>...", "image":
"<IMAGE_NAME>"}
```

## 3.3 HANDLING UNEVEN DATA

We observed that certain classes, such as tanks and artillery, appeared much more frequently in our dataset compared to others like APCs and helicopters. To prevent model bias, we applied a technique called class-component sampling during training Buda et al. (2018). To further balance the dataset, we augmented the underrepresented categories using simple data augmentation techniques such as random cropping, flipping, and rotation Perez & Wang (2017).

## 3.4 MODEL ARCHITECTURE

Florence 2 is a unified multimodal architecture designed to process and align both visual and textual inputs through a dual-stream encoding and decoding mechanism. The model begins with separate pathways for vision and language. The vision encoder leverages a Vision Transformer (ViT) to process input images into contextualized embeddings, while the language encoder employs a transformer-based structure to encode input text into rich token-level representations. Both encoders operate independently but are synchronized through a shared latent space that enables effective multimodal understanding.

The outputs of the encoders are fed into a unified decoder framework featuring two decoding paths: a standard decoder and an interactive decoder (Decoder IC). These decoders utilize a search mechanism for task conditioning and contextual relevance. Embedded features from both modalities are fused through cross-attention layers to generate meaningful outputs, which are then passed through a linear projection and post-processing modules including a post prompt and finalizer. This design allows Florence 2 to handle a wide range of vision-language tasks, offering modularity, scalability, and high adaptability across diverse applications Krizhevsky et al. (2017); GitHub.

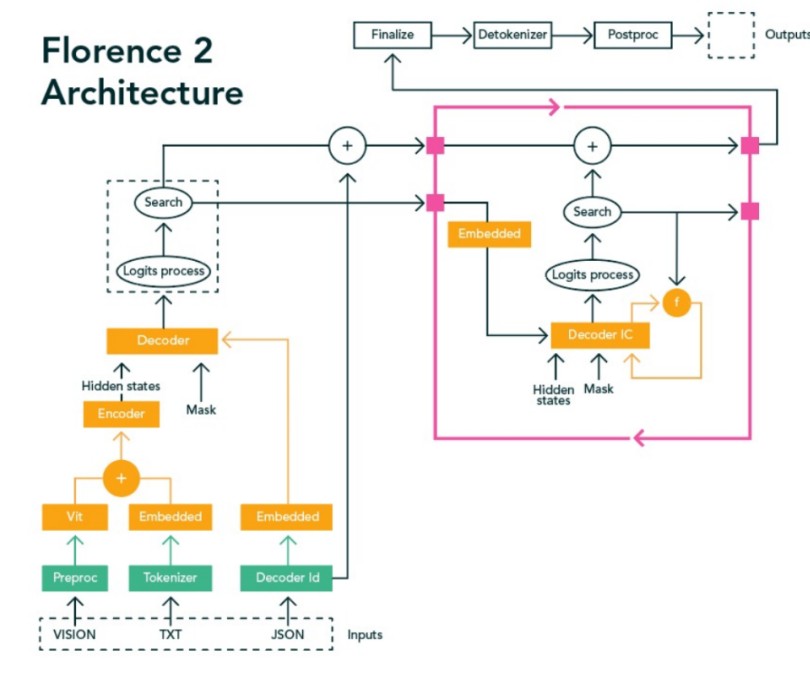

Figure 2: Florence-2 architecture overview, highlighting dual- stream encoders (vision and language) and unified decoders with cross-attention for multimodal fusion.

### 3.5 MODEL TUNING

We started with the official pre-educated weights of Florence 2 and then fine-tuned the network using our specially curated dataset.

**Loss function:** We used an IoU-based loss function to classify objects and predict their bounding boxes, as it's proven to be more effective for localization tasks than traditional L1/L2 losses Rezatofighi et al. (2019).

**Major settings:** We set the learning rate using a cosine decay schedule, beginning with a base value of $5\times10^{-5}$, and used a batch size of 16. We also implemented early stopping to prevent overfitting, training was halted if the validation accuracy stopped improving Loshchilov & Hutter (2016).

**Training Time:** The model was trained over 10 epochs, and we always saved the checkpoint that performed best on our validation set.

### 3.6 VERIFICATION AND EVALUATION

We set aside 20% of our dataset for validation purposes. After each training cycle, we carefully monitored critical performance metrics such as accuracy and recall to evaluate the model's effectiveness Goodfellow et al. (2016). In addition to numerical evaluation, we also manually inspected a sample of the model's predictions to ensure it had learned to recognize and distinguish real-world scenarios accurately, especially in the case of visually complex or overlapping categories Zhang et al. (2016).

This combination of systematic validation, both quantitative and qualitative, gives us confidence that the model can perform reliably in practical deployments, particularly in mission-critical defense applications where precision is essential.

## 4 EXPERIMENTAL RESULTS

To validate the performance of our fine-tuned Florence-2 model for detecting military artillery, we conducted a comprehensive evaluation using the curated dataset of 7,021 validation images. We

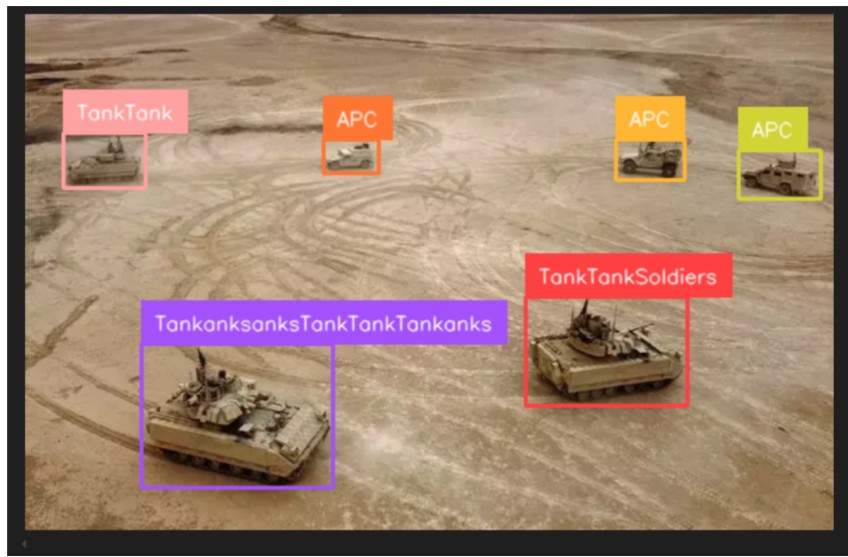

Figure 3: Sample predictions from the fine-tuned Florence-2 model. The model successfully detects camouflaged and partially occluded objects in battlefield-like scenarios.

utilized widely accepted object detection metrics such as Precision, Recall, F1 Score, Accuracy, Average IoU, and mAP@0.5 to assess both detection quality and robustness in realistic battlefield scenarios.

## 4.1 QUANTITATIVE METRICS

The evaluation yielded the following high-performance results:

Table 1: Performance metrics of the fine-tuned Florence-2 model on the validation set.

| Metric | Value |
| --- | --- |
| Precision | 98.95% |
| Recall | 99.33% |
| F1 Score | 99.14% |
| Accuracy | 98.29% |
| Average IoU | 85.29% |

These metrics indicate the model is highly effective at detecting relevant military targets such as tanks, aircraft, and artillery units with minimal false positives or missed detections.

## 4.2 COMPARATIVE PERFORMANCE: FLORENCE-2 VS. YOLOV5 VARIANTS

To assess how our model compares to existing state-of-the-art architectures, we benchmarked Florence-2 against various YOLOv5-based models using the mAP@0.5 metric, a widely used indicator of detection quality.

mAP@0.5 (mean Average Precision at IoU $\geq 0.5$) measures the overall detection accuracy of a model. A prediction is considered correct if its IoU with the ground truth is at least 0.5. The final score is the mean of the Average Precisions (AP) across all classes:

$$\text{mAP@0.5} = \frac{1}{N} \sum_{i=1}^{N} \text{AP}_i, \quad \text{where IoU} \geq 0.5$$

Table 2: Comparative performance of Florence-2 and YOLOv5-based models on artillery detection using mAP@0.5.

| Model | mAP@0.5 |
| --- | --- |
| YOLOv5s | 96.5 |
| YOLOv5s+Stem | 95.9 |
| YOLOv5s+MNtV3 | 96.8 |
| YOLOv5s+Stem+MNtV3 | 96.6 |
| YOLOv5s+Stem+MNtV3-CBAM | 97.3 |
| SMCA-YOLOv5 | 97.8 |
| **Fine-tuned Florence-2 (Ours)** | **98.29** |

As seen in Table 2, the fine-tuned Florence-2 model achieves the highest mAP@0.5 score of 98.29%, clearly outperforming all other YOLO-based architectures, including YOLOv5s, YOLOv5s+MNtV3, and even SMCA-YOLOv5. This demonstrates not only superior detection accuracy but also the model's robustness and reliability in complex military scenarios where precise object identification is critical.

### 4.3 LOSS TRENDS

From the loss plots, we ought to honestly have a look at how the version, as also described in general training dynamics by Goodfellow et al. Goodfellow et al. (2016), progressively discovered better with each epoch: The schooling loss showed an easy and steady decline, beginning above 2.1 and ultimately flattening underneath 0.85 via the tenth epoch. The validation loss, in a way that aligns with trends noted in Zhang et al. Zhang et al. (2016), initially fluctuated slightly, peaking around epoch three, but gradually reduced over time, settling near 1.13 within the final stages. This kind of behaviour is expected while working with a dataset that has real-world noise Zhang et al. (2016), and the drop in both training and validation loss over time, as discussed in general deep learning literature Goodfellow et al. (2016), is a great signal that the version is generalizing well.

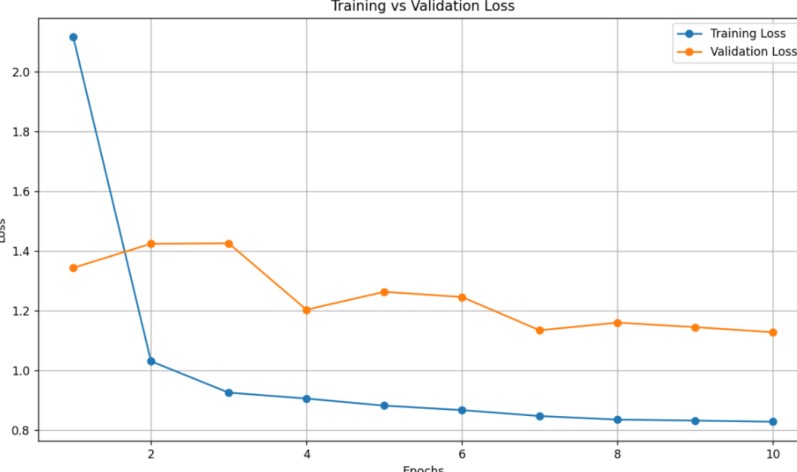

Figure 4: Training and validation loss trends for Florence-2 fine- tuning.

### 4.4 TRAINING AND VALIDATION TIMINGS

We additionally tracked how lengthy each epoch took: On average, every training epoch took around 240–246 seconds. Validation changed into barely faster, averaging approximately one hundred twenty five–129 seconds in step with epoch. These timings helped us become aware of the sweet

spot for training without overloading sources, specifically if we need to later deploy the model on part gadgets.

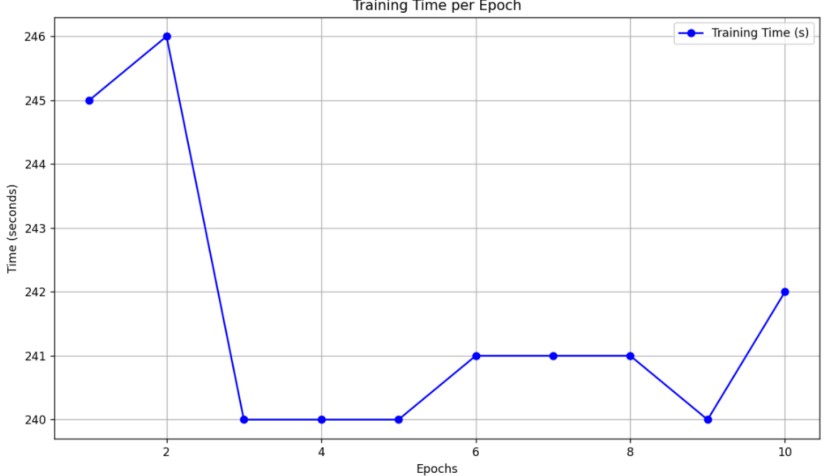

Figure 5: Training time per epoch.

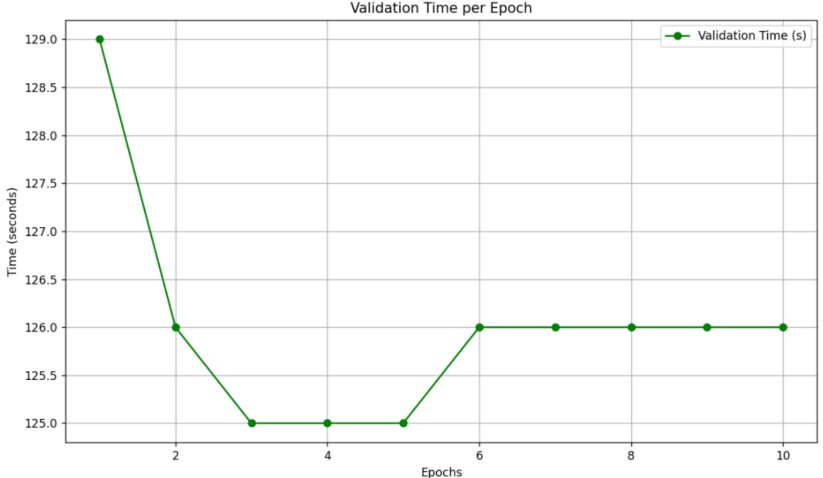

Figure 6: Validation time per epoch.

## 4.5 OVERALL INSIGHTS

The version's overall performance suggests that the Florence-2 architecture, as also indicated in recent work on foundation vision models by Yuan et al. Xiao et al. (2023), whilst fine-tuned on a well-curated and annotated dataset, is exceedingly capable of knowledge and detecting various army utilities like tanks, IFVs, helicopters, and more even in real-world, noisy, and camouflaged environments.

The mixture of high F1 score and strong IoU, consistent with benchmarks in object detection literature Padilla et al. (2021), proves that the predictions were not simply accurate but also well localized. These results, echoing the findings in Xiao et al. (2023), give us confidence in its use for battlefield surveillance and threat identification in real-time systems.

## 5 CONCLUSION

In this work, we proposed a practical and fine-tuned approach for the detection of military vehicles and artillery units using Microsoft's Florence-2 vision-language model. The core idea was to go beyond synthetic or staged datasets and instead focus on real-world battlefield scenarios where detection accuracy genuinely matters, a concern also emphasized in prior works on combat-relevant computer vision systems Jacob et al. (2023).

Our dataset, curated from real-world footage and articles, was refined carefully to represent actual deployment conditions, including blurry, low-light, and camouflaged instances, as often seen in military imaging challenges Bahi et al. (2024). The preprocessing pipeline, combined with class-aware augmentation techniques Perez & Wang (2017), helped address the inherent imbalance present across different types of equipment.

### 5.1 OVERVIEW OF WHOLE PROCESS

Through rigorous fine-tuning and validation, our model achieved an F1 score of 99.14% and an average IoU of 85.29%, which clearly demonstrate the robustness and reliability of our approach. The smooth convergence in training and validation loss, consistent with deep learning best practices Goodfellow et al. (2016), further assures the model's generalisation capabilities.

We believe that this work contributes not only a high-performing model, but also a replicable methodology that others working in the security and defence space can build upon Zhang et al. (2016). Going ahead, we aim to explore deployment on edge devices (e.g., NVIDIA Jetson, Raspberry Pi), extend the model to identify enemy camouflage and advanced artillery types, and make it more robust to adversarial conditions like dust, smoke, and motion blur conditions frequently cited in battlefield imaging research Bahi et al. (2024).

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
