# OpenReview forum: "Military Object Detection Using a Fine-Tuned Florence Vision Model"
_ICLR.cc/2026/Conference — ICLR 2026 Conference Desk Rejected Submission_

### Official Review · Reviewer_GBpc · 2025-10-21

**Soundness:** 3
**Presentation:** 1
**Contribution:** 1
**Rating:** 2
**Confidence:** 5

**Summary:**

This paper creates a new military dataset with 7000 cleaned images, that includes military vehicles and personnel. The authors propose fine-tuning Microsoft’s Florence-2 Large model on this dataset and show that the fine-tuned model outperforms a fine-tuned YOLOv5 model. The authors employ class-component sampling and simple data augmentation techniques to overcome the class imbalance problem.

**Strengths:**

Overall, the contribution of 7000 cleaned images is good. Furthermore, the analysis of the Florence-2 model indicates that it can achieve very high mAP@0.5 on this dataset. At this IoU threshold, it is clear that the fine-tuned Florence-2 model outperforms many YOLOv5 models by a reasonable margin.

The quantitative analysis and metrics used to compare the Florence and YOLOv5 models are sound (although not sufficient) and clearly presented.

**Weaknesses:**

1. Overall my main issue is the writing style of the paper. There are language tone issues throughout the paper that seriously detract from its credibility. A few examples:

line 040-041: "Most datasets available online show tanks and jets in display conditions,
clean, centered, and fully visible. But real war doesn’t happen in exhibition grounds..."

line 054-056: "Training such a model was no easy task. Camouflage, shadows, damaged vehicles, and background
clutter made it tough, but that was the point. We wanted the model to be tough too..."

line 065: "we went the extra mile to..."

line 120-121: "Quality Check: We tossed out any photographs that had been remarkably blurry or distorted – the
type even someone would wear to make sense of..."

line 291: "From the loss plots, we ought to honestly have a look at how the version..."

####################################################################################################
Dataset contribution:

2. The dataset described in this paper includes 7000 cleaned and annotated images. This has the potential to be a strong contribution, but currently there is not enough detail provided to have a clear understanding of the contribution of this dataset. Currently the reader does not have enough detail to ascertain how easy or hard this dataset is, and whether a model that performs well on it would be suitable for their own tasks. The description of the dataset would benefit with examples.

2a. For example on line 120-121 the authors state: "We tossed out any photographs that had been remarkably blurry or distorted – the type even someone would wear to make sense of." Visualisation of these types of images would be helpful.

2b. On line 78-80 the authors state: "Several images turned out to be too blurry or unclear, in fact, some were hard to recognize even for the human eye. Acknowledging these limitations and to stay true to quality standards, we applied a human baseline filter and truncated the dataset accordingly." This is reasonable. Can we have a visualisation of the types of images that are and aren't included

The SAM [1] paper (https://arxiv.org/pdf/2304.02643) has good example of exact annotation strategy for reference

3. On line 141 the authors state: "To further balance the dataset, we augmented the underrepresented categories using simple data augmentation techniques such as random cropping, flipping, and rotation..." More discussion is required here. By how much were the images upsampled? What was the distribution of the classes after additional augmentation and class-component sampling? More discussion is needed around augmentation, especially for dealing with underrepresented categories. Furthermore, the YOLO architectures are quite well established. What repository was used as no citation is provided? More discussion is required here as these repositories have lots of training parameters, including mosaic training by default. Was mosaic training used in addition to what was reported? Did training the Florence model receive the exact same data augmentation during training?

####################################################################################################

Model analysis:

4. In order to conclude that Florence is a suitable model, more quantitative analysis is required. Florence vs YOLOv5 is not a sufficient benchmark, as YOLOv5 is now five years old, and SMCA-YOLOv5 is at least three years old. YOLOv12 now exists. Comparisons to more modern architectures need to be made, including non-YOLO architectures. This paper makes no comparisons to other transformer architectures, for instance.

4a. On line 044-045 the authors state: "After comparing several leading models, we selected Microsoft’s Florence-2 for its powerful combination of image understanding and language processing..." If there was preliminary round of experimentation that lead to selection of Florence, with either quantitative or qualitative analysis, this should be included. Which other models were shortlisted? Maybe we don't need to compare additional fine-tuned models if there is sound motivation to discard them after an initial round of evaluation.

5. No citations are provided for the other YOLO architectures.

6. There is no indication under what condition Florence performs better than YOLO. Only the mAP metric at threshold 0.5 is considered, what about 0.5:0.95? This is a standard metric to evaluate under. COCO metrics could be easily incorporated into the quantitative analysis to investigate performance at different instance sizes. We need qualitative analysis of the performance of Florence compared to different models. Success and failure cases should be visualised, including examples where Florence performed well and another model did not, and vice-versa.

7. There is no attempt to explain why Florence performs better than other architectures. Is it because of transformer architecture? Is it because it is a vision-language model? Is there a more nuanced explanation? Without comparisons to more types of architectures it is difficult to properly explain why Florence is well-suited to the task. The paper would benefit from qualitative analysis of what features the fine-tuned Florence model is paying attention to compared to other models.

####################################################################################################
A couple of errors I noticed that did not impact my opinion of the paper:

line 121: "the type even someone would WEAR to make sense of..."

line 321: "We additionally tracked how LENGTHY each epoch took"

####################################################################################################
References:

[1] A. Kirillov et al., ‘Segment Anything’, in IEEE/CVF International Conference on Computer Vision, ICCV 2023, Paris, France, October 1-6, 2023, 2023, pp. 3992–4003.

**Questions:**

What kind of images were are were not included in the dataset? Provide visual examples

Provide more detail about the class-component sampling and augmentation. Was mosaic training used? Did all YOLO architectures and Florence receive exact same training-time image augmentation? What was the distribution of classes before and after balancing strategies?

Was the 20% train/validation split random or in any way class-aware?

Why were only Florence-2 and YOLOv5 architectures chosen? Are better models available? Under what specific conditions is Florence better than YOLOv5 and vice-versa? Are there any general patterns?

---

### Official Review · Reviewer_9iLB · 2025-10-23

**Soundness:** 1
**Presentation:** 1
**Contribution:** 1
**Rating:** 0
**Confidence:** 4

**Summary:**

The paper fine-tunes Florence-2 for detecting military objects in real-world conditions. Using a self-curated dataset of about 7,000 battlefield simulations and surveillance images, the model achieves strong results and outperforms YOLOv5 variants, showing potential for defense applications such as automated surveillance and threat detection.

**Strengths:**

1. The work addresses a practical and difficult real-world setting, focusing on realistic, non-staged military imagery that is underexplored in public benchmarks.

2. The dataset construction process is carefully documented, including annotation procedures and class balancing through augmentation.

**Weaknesses:**

1. The paper lacks sufficient methodological novelty, the main contribution is dataset curation and fine-tuning, rather than architectural or algorithmic innovation.

2. The dataset collection and annotation process rely on online and potentially unverified sources, raising reproducibility and ethical concerns about data provenance.

3. The evaluation focuses heavily on single-dataset metrics; there is no test on unseen or cross-domain datasets to validate generalization.

4. Many sections use general explanations and do not provide implementation specifics (e.g., optimizer type, compute resources, data license), limiting reproducibility.

5. This paper lacks analysis and interpretation of relevant work and fields.

6. I believe the writing quality and figure quality of this paper do not meet the standards of ICLR.

**Questions:**

Same as weaknesses section.

**Details Of Ethics Concerns:**

This work focuses on military task, and I am not sure whether the application (positive, negative?) and data usage of this paper requires more discussion about ethics.

---

### Official Review · Reviewer_5B5S · 2025-10-28

**Soundness:** 2
**Presentation:** 2
**Contribution:** 2
**Rating:** 2
**Confidence:** 1

**Summary:**

This work proposes using a VLM-based Foundation model (Florence-2) to fine-tune for the object detection task. The paper discusses the lack of variability in the current datasets used to train and evaluate object detection tasks in the military context. The main contribution of the paper is towards building a new dataset, which consists of 8 major classes consisting of unique 7000 images, for the purpose of training an Object detection tasks

**Strengths:**

Strength:

1. The paper rightly focuses/identifies the shortcomings from a data point of view in the existing evaluation/training paradigm for military object detection.
2. The paper attempts to collect real images from real video footage and real war-field images and builds on that to capture messy/varied and real-looking images for training the OD task
3. Leverages on the tools such as CVAT and Roboflow for robust image labeling and creating a trainable set of ~7K images

**Weaknesses:**

Weakness:

1. There is no technical/architectural novelty presented/discussed in this work, which suggests improving the current architecture of the existing models to better leverage the new dataset for OD.
2. Paper talks about data normalization, but fails to discuss in detail/show scenarios related to: How does the model deal with poor/low-lighting scenarios? And what are the improvements seen in that specific scenario
3. There is no serious discussion/ablation in the work to prove how the variability in the dataset, captured from different sources, are "taken care of" while training the model?
4. What kind of normalization? What are other ways explored, and do we have related ablation experiments showing the efficacy of the approach selected in this paper?
5. Model Architecture Section only talks about the Training Model architecture of the Florence-2 model, nothing is mentioned about what trainable module introduced by this work to train for an OD task
6. No information / and related ablation for truncated object classes and evaluation/improvement on those object instances only?
7. No Information/and related ablation for bounding boxes based on the size of the BB, by categorizing them as small/medium/large and estimating AP based on the that.

**Questions:**

Please see the Weakness section

---

### Official Review · Reviewer_Ndrj · 2025-10-30

**Soundness:** 2
**Presentation:** 1
**Contribution:** 2
**Rating:** 0
**Confidence:** 5

**Summary:**

This work creates a militarily object detection dataset replicating real battlefield conditions, including over 7,000 images sourced from exercises and surveillance footage. The Florence-2 large vision-language model was fine-tuned on this dataset and demonstrated better object detection performance compared to YOLOv5 series models in comparative experiments.

**Strengths:**

The authors collected over 7,000 real-world battlefield images from sources like military exercises and surveillance footage, instead of using existing standard datasets captured in ideal environments. These images include complex scenarios such as camouflage, partial occlusion, low lighting, and blur, providing a valuable data resource for military object detection research in the future.

**Weaknesses:**

The shortcomings of this paper are quite apparent, as listed below:
1.This paper is weak in motivation description. Regarding the dataset, the motivation for creating it is only briefly mentioned with a few qualitative terms. Readers cannot understand the following issues: What are the specific distinctions between this dataset and those used in prior work? What are the concrete difficulties this new dataset adds to this task? Why the Florence-2 model was specifically chosen for this more challenging task and where does its suitability lie? The deficiency in motivation description makes this work appear more like a simple model application report rather than a deep methodological study.
2.The paper demonstrates extremely weak innovation. The core contribution in model design is merely fine-tuning the pre-existing Florence-2 model. Firstly, highlighting "fine-tuned" as a key methodological aspect in the title is arguably inappropriate. Secondly, beyond the dataset, it is difficult to identify any other applied valuable innovations in this work.
3.The workload appears insufficient. The model design is too simple, which directly leads to a lack of substantive content in the Experimental section. For instance, sections 4.3 and 4.4 read more like routine recordings of the training process rather than providing in-depth analysis linking these observations to generalization capabilities or architectural properties.
4.The comparative experiments are inadequate. The benchmarks used are outdated, as the selected models for comparison are all YOLOv5 and its variants. Furthermore, the comparison experiment only uses mAP@0.5 to measure the result, which appears to be too simple.
5.The paper contains numerous noticeable formatting issues, primarily related to citation formatting.
6.The Conclusion section appears unfinished, lacking a detailed discussion of its deficiency and potential further improvement.

**Questions:**

It is recommended that the authors refine the motivation section to improve overall narrative coherence. Additionally, incorporating comparisons with more recent object detection benchmarks would make the experiment results more convincing.

---

### Note · Program_Chairs · 2026-01-17
**Submission Desk Rejected by Program Chairs**

The following references in this submission do not refer to real documents and/or have major errors in bibliographic information:

 Amine Bahi, Lamia Rebbah, and Samir Rebai. Object detection in low light environments. Procedia Comput. Sci., 246:3381-3389, 2024.